# Research on Solid Shell Growth during Continuous Steel Casting

**DOI:** 10.3390/ma16155302

**Published:** 2023-07-28

**Authors:** Marek Velička, René Pyszko, Mario Machů, Jiří Burda, Tomáš Kubín, Hana Ovčačíková, David Rigo

**Affiliations:** 1Department of Thermal Engineering, Faculty of Materials Science and Technology, VSB-Technical University of Ostrava, 17. listopadu 2172/15, 708 00 Ostrava, Czech Republic; rene.pyszko@vsb.cz (R.P.); mario.machu@vsb.cz (M.M.); jiri.burda@vsb.cz (J.B.); hana.ovcacikova@vsb.cz (H.O.); david.rigo@vsb.cz (D.R.); 2Department of Machine and Industrial Design, Faculty of Mechanical Engineering, VSB-Technical University of Ostrava, 17. listopadu 2172/15, 708 00 Ostrava, Czech Republic; tomas.kubin@vsb.cz

**Keywords:** continuous casting of steel, shell thickness, laser scanner, breakout, modelling

## Abstract

The continuous steel casting process must simultaneously meet the requirements for production performance, quality and safety against breakouts. Knowing the thickness of the solidified shell, particularly at the exit of the mould, is useful for the casting process control and breakout prevention. Shell thickness is difficult to measure during casting; in practice, it is predicted by indirect methods and models. But after undesired rupture of the shell and leakage of the liquid steel, it is possible to measure the shell thickness directly. This article is focused on the problem of the growth and measurement of the solid shell obtained after the breakout of a round block with a diameter of 410 mm. An original methodology was developed in which a surface mesh of points was created from the individual scanned parts of the block using a 3D laser scanner. Research has shown differences of up to 6 mm between the maximum and minimum shell thickness at the mould exit. A regression function of the average shell thickness on time was found. The results of the real shell growth were further used for the verification of the original numerical model of cooling and solidification of the round block.

## 1. Introduction

Continuous casting is a highly productive and efficient method of converting molten metal into a solid shaped semi-finished product. Almost all liquid steel produced today is processed in this way. Compared to classical casting in stationary moulds, in continuous casting, the steel is only present in the mould for a limited time and proceeds further through the casting machine. During this time, the steel does not solidify in the entire cross-section, but only a solid shell is formed and the liquid core remains. The area of solid shell formation in the mould is called the primary cooling zone. This partially solidified steel leaves the mould and enters the secondary cooling zone, where the steel is cooled by water spraying and the shell is supported by the rolls.

The thickness of the solid shell leaving the mould must be sufficient to withstand not only the static pressure of the liquid metal but also mechanical and thermal stress. At the same time, from the point of view of quality, the thickness of the shell should be uniform around the perimeter of the cross-section and should grow evenly lengthwise. If these conditions are not met, there is a risk of cracks forming and, in an extreme case, breaking the shell and leaking liquid steel, which is called breakout.

Improper growth of the solid shell in the mould may not immediately lead to a breakout, but the beginnings of defects formed in the mould can further develop in the secondary zone due to incorrect cooling. This is when a large thermal and mechanical stress arises and the strength of the material is overcome. These are not only external defects that are visible, but also internal defects that are hidden at first glance and can cause major problems in further technological procedures, such as in hot or cold metal forming [1].

Casting technology must meet two basic requirements that mostly work against each other: namely, to ensure sufficient performance and at the same time to maintain the prescribed quality of the cast production with sufficient prevention against breakouts. From this point of view, an important indicator is the intensity of heat transfer from the strand to the mould, which affects the rate of growth of the solid shell [2]. The working conditions in the primary cooling zone have a significant influence on the surface quality of the product and the formation of the uniform and sufficiently thick defect-free solid shell. It is known that the first causes of defects are formed already in the mould.

The very understanding of thermal processes during continuous steel casting is important as it allows choosing the right casting and cooling parameters and thus optimizing thermal processes during continuous casting, predicting the occurrence of defects and minimizing the risk of breakouts [3].

The liquid steel that comes into contact with the copper wall of the mould is intensively cooled and begins to solidify on the surface of the wall [4]. The shell begins to form in the meniscus, i.e., in the place where the surface of the liquid metal meets the surface of the mould. The shell continues to grow with increasing distance from the meniscus. The process of formation of the first shell in the area of the meniscus is complex and is related to a number of physical phenomena [5].

Some research focuses on the growth of the shell just below the meniscus [6], where the rate of cooling and solidification was estimated on the basis of experimental results of solidified shell thickness, heat flux in the mould and dendrite arm spacing in the solidified structure beneath the surface of the blank. As a result, it was found that there is a delaying period of solidification growth at the beginning, until the shell grows up to about 1 mm thick. After that, for a limited time, it grows approximately in linear relation to the square root of solidification time.

Efforts to prevent solid shell cracks and breakouts have recently led to the rapid development of operational systems for predicting the thickness of the solid shell, which is often combined with the continuous monitoring of thermal and mechanical quantities in the mould [7]. In connection with this, a number of theoretical–experimental research studies have been carried out aimed at gaining knowledge about the behaviour of the continuous casting process and finding dependencies between thermal–mechanical quantities in the mould, the formation of the solid shell and casting parameters [8,9,10,11].

Breakout prediction systems based on the indication of a crack in the shell are usually based on the measurement of temperatures and mechanical quantities in the mould. These systems indicate an already developed crack and are based on mould instrumentation and prediction algorithms that work with measured quantities, especially temperatures in the mould wall. In addition, modern casting machines are equipped with systems that evaluate the risk of breakout based on the shell thickness prediction. These systems have a wider use, as they can also share information with systems for a dynamic control of the cooling intensity in the secondary zone [12]. Since the thickness of the shell is not a directly measurable quantity, these systems are mostly based on mathematical models [13,14]. Indirect methods such as artificial intelligence can also be used, e.g., in combination with the use of special sensors such as laser vibrometer [15]. The new logic-based mould breakout prediction systems have been developed for continuous casting machines. One of such prediction systems not only detects sticker breakouts but also breakouts that take place due to variations in the casting speed, mould level fluctuation and taper/mould problems [16].

An exact mathematical description of the thermal processes during the continuous casting of steel is difficult to compile, because many different influences affect the cooling and solidification of the blank [17]. Therefore, it is necessary to search for the quantities that have the greatest influence on the blank solidification [18].

There are several ways for creating mathematical models of the solid shell growth. Models can be based, first, on an analytical solution based on physical laws, second, on a statistical model built on measured data, and the third option is the use of numerical methods based on differential equations, in particular the finite difference method [19,20,21], the finite volume method [22] or finite element method [23,24,25,26,27].

In operational conditions, thermal models are used to predict temperatures of the blank and the solid shell thickness. The results of such prediction models, which work online in real time, can be also used as input data in breakout prediction systems [28]. Currently, sophisticated numerical models can be used to perform not only thermal calculations but also calculations of stress, structure and chemical heterogeneity, including segregation prediction [29,30]. Due to the high computational complexity, calculations of such quantities are performed offline for the purpose of research and development of casting new steel grades or formats. Commercial programs such as ANSYS [31], ProCAST, etc. can be used to model the process of steel cooling and solidification. The main obstacle to the wider use of these computing systems in operational conditions is the complexity of physico-chemical processes, the demand for computing power and time and the dependence on detailed boundary conditions that are usually unknown. These methods are not the subject of this article, which focuses on the operational methods of shell thickness prediction using regression equations and a fast online numerical thermal model and its verification by comparing the results with measured thickness of the real solid shell obtained after a breakout.

The presented article aims to discuss the process of continuous casting, highlight the importance of maintaining quality and uniformity in the solid shell formation, address the risks of defects and breakouts, emphasize the understanding of thermal processes, and explore the use of mathematical models and predictive systems for thickness prediction.

## 2. Modelling of the Solid Shell Growth

The thickness of the solid shell is determined by general physical laws and depends on a number of parameters. The growth of the solidified shell is influenced by the casting speed, the thermophysical properties of the cast steel and the intensity of heat transfer [32]. Heat removal from the steel is limited both by internal heat transfer in the steel and by external heat transfer at the boundary between shell–mould and mould–cooling water [33]. Internal heat transport is mainly influenced by the intensity of convection in the liquid steel and thermal conductivity. External heat transfer in the mould is most intense in the upper part of the mould, where the steel is in good contact with the mould wall. In the lower part of the mould, the heat transfer is reduced by the thermal resistance of the gaseous gap, which forms due to the thermal shrinkage of the solidifying steel [34]. Even reheating of the solid shell may occur in the lower part of the mould [35]. Static pressure acts on the solid shell against shrinkage. Additional thermal resistance occurs in the lubricant layer between the shell and the mould wall. Other influences enter the process, which may have even a periodic or random character and which cause unevenness of the shell both around the perimeter and along the blank. Earlier studies have shown that the shell thickness around the circumference of the billet in low-carbon steels is significantly more uneven compared to steels with a higher carbon content [36].

### 2.1. Modelling Methods of the Thickness of the Solid Shell

Analytical and empirical models of the thickness of the solid shell provide only indicative and approximate values, as they cannot reflect detailed conditions of heat removal. Solidification conditions depend not only on the size and shape of the cross-section of the blank but also on other casting parameters that differ for each heat and even during the heat itself due to their technological and temporal variability. This is mainly about the temperature and chemical composition of the steel, the geometry and wear of the mould, cooling intensity of the mould and other, sometimes unknown and random influences.

The growth of the solid shell very closely below the meniscus, as well as the formation of the solid phase in the classical casting of steel in stationary moulds, can be described by the Neumann parabolic law, where the thickness of the solid phase is a linear function of the square root of the solidification time.
(1)$$\xi =K\cdot \tau^{0.5}\;\;\;(\mathrm{mm})$$
where *K* is the solidification coefficient (mm·min^0.5^) and *τ* is time (min).

This formula is often modified into various forms, e.g., [32,37]
(2)$$\xi =K\cdot \sqrt{\frac{L}{w}}\;\;\;(\mathrm{mm})$$
where *L* is the distance from the meniscus (m), and *w* is the casting speed (m·min^−1^).

However, many experimental measurements have already shown that the growth of the shell does not proceed exactly according to the parabolic law [38]. In the case of continuous casting, which is a much more complicated process dependent on many factors, the shell thickness at a greater depth below the meniscus can be described by empirical regression formulas in the form of the power function of time
(3)$$\xi =K\cdot \tau^{n}\;\;\;(\mathrm{mm})$$
where *K*, *n* are generally constants dependent on the type of the caster and the steel parameters, and *τ* is the time it takes for the steel element to pass from the surface to the given position of the mould.

The parameters of this function depend on the dimensions and shape of the blank, the overheating of the steel above the liquidus temperature, the thermophysical properties of the steel, the liquidus and solidus temperatures of the steel, the intensity of heat removal etc.

A number of specific empirical formulas can be found in the literature. Tsuneoka [39] states the relationship
(4)$$\xi =1.475\cdot \tau^{0.66}\;\;\;(\mathrm{mm})$$
where *τ* is time (s).

To determine the thickness of the solid shell, the authors Janik and Dyja [40] used the Chipman–Fondersmith relationship which has the form after conversion to metric units
(5)$$\xi =29.51\cdot \sqrt{\tau }-3.048\;\;\;(\mathrm{mm})$$
where *τ* is time (min).

A similar formula was used by researchers at AGH Krakow [27] which also requires to enter time in minutes
(6)$$\xi =22.86\cdot \sqrt{\tau }-3.05\;\;\;(\mathrm{mm})$$

The thickness of the shell can also be determined, based on the well-known Neumann relation
(7)$$\xi =K\cdot \sqrt{\tau }$$
where *τ* is time (min) and *K* is a function [13]
(8)K=13.624 lnC−0.0572∆T−90.89
where *C* is a constant (kW·m^2^·s^0.5^) whose value depends on a format of the actual mould and Δ*T* (K) is steel overheating above the liquidus temperature.

The process solidification in the mould also depends on steel chemical composition, especially on the carbon content. This process is characterized by a liquidus temperature *T*_liq_, which represents the beginning of solidification, and a solidus temperature *T*_sol_, at which the solidification ends. Between these two temperatures (in the so-called “mushy” zone), there is a certain proportion of the solid phase *f_s_*, which is characterized by a value between 0 and 1.

Multiple models of solid-phase formation between liquidus and solidus temperatures are used [25], for example
(9)$$f_{s}=\left(\frac{1}{1-k_{0}}\right)\cdot \frac{T_{liq}-T}{T_{sol}-T}\;\;\;(1)$$

The thickness of the solid shell is then usually defined by a certain *f_s_* value at a particular point. The solid phase represents a contractual value of *f_s_* at which the properties of the mushy phase are already approaching the fully solidified phase.

The models mentioned above can predict an average shell thickness around the circumference of the blank cross-section. However, in the continuous casting process, the steel does not solidify evenly around the perimeter. The thickness of the shell at a given horizontal level usually varies by tens of percent; in many cases, the differences are even greater. Thus, a breakout can occur even if the calculated average thickness of the solid shell at the end of the mould is sufficient [41].

A more accurate description of the process of heat removal and steel solidification is possible using differential equations of heat conduction, possibly in combination with equations of flowing.

The formation of the solid shell is related to the kinetics of the temperature field of the blank. The non-stationary temperature field without considering the steel flow is described by the Fourier partial differential equation
(10)$$\frac{\partial t}{\partial \tau }=a\cdot \nabla^{2}t+\frac{q_{V}}{c_{p}\cdot \rho }\;\;\;(K\cdot s^{-1})$$
where *t* is the steel temperature (K), *τ* is time (s), *a* is temperature diffusivity (m^2^·s^−1^), ∇^2^ is the Laplace operator (m^−2^), *c_p_* is the specific heat capacity (J·kg^−1^ K^−1^), *ρ* is density (kg·m^−3^), and *q_v_* is the intensity of the internal volumetric heat source (W·m^−3^).

The Fourier–Kirchhoff equation can be used to describe the temperature field with the flow of liquid steel and the movement of the solid shell
(11)$$\frac{Dt}{d\tau }=a\cdot \nabla^{2}t+\frac{q_{V}}{c_{p}\cdot \rho }\;\;\;(K\cdot s^{-1})$$
where the expression on the left side of Equation (11) represents the substantial derivative of temperature. In the specific case of a round blank, it is more appropriate to solve the equation in the cylindrical coordinate system, and the substantial derivative has the form
(12)$$\frac{Dt}{d\tau }=\frac{\partial t}{\partial \tau }+w_{r}\cdot \frac{\partial t}{\partial r}+\frac{w_{\varphi }}{r}\cdot \frac{\partial t}{\partial \varphi }+w_{z}\cdot \frac{\partial t}{\partial z}\;\;\;(K\cdot s^{-1})$$
where *w_r_*, *w_φ_*, and *w_z_* are the velocity components in the directions of cylindrical coordinates (m·s^−1^).

From Equation (11), with regard to Equation (12), it is clear that the temperature of the flowing liquid steel is a function of both the independent variables *r*, *ϕ*, z, *τ* and also the velocity components *w_r_*, *w_φ_*, and *w_z_*.

The Fourier–Kirchhoff equation must therefore be solved together with three Navier–Stokes equations of motion, which can be written in vector form
(13)$$\frac{Dw}{d\tau }=A-\frac{1}{\rho }\cdot grad\;p+\nu \cdot \nabla^{2}w\;\;\;(m\cdot s^{-2})$$
where *p* is pressure (Pa), ***A*** is acceleration (m·s^−2^), and *ν* is the kinematic viscosity of liquid steel (m^2^·s^−1^).

The pressure gradient for a cylindrical system can be expressed by the equation
(14)$$grad\;p=\frac{\partial p}{\partial r}+\frac{1}{r}\cdot \frac{\partial p}{\partial \varphi }+\frac{\partial p}{\partial z}\;\;\;(\mathrm{Pa}\cdot m^{-1})$$

The differential description of the temperature field is the basis of numerical models. Thermal models used in operational prediction systems usually do not address the spatial flow of liquid steel but are limited to longitudinal motion only. Equation (8) is thus reduced to a simpler form
(15)$$\frac{Dt}{d\tau }=\frac{\partial t}{\partial \tau }+w_{z}\cdot \frac{\partial t}{\partial z}\;\;\;(K\cdot s^{-1})$$

The velocity *w_z_* is equal to the casting speed, and the need to solve Equations (14) and (15) is therefore eliminated. A possible attempt to use a more detailed mathematical description in practice runs into unknown boundary conditions.

Numerical models will generally allow more accurate solutions than analytical models. A prerequisite for obtaining the correct solution of differential equations is the knowledge of the boundary conditions.

The boundary condition of the III^rd^ kind is usually specified in the mould, i.e., heat flux. There are many empirical formulas in the literature to calculate heat flux in the primary cooling zone [42]. They are mostly exponential functions of the longitudinal coordinate *z* and the casting speed *w*, for example [43]
(16)$$q=A\cdot w^{0.56}\cdot exp(-\upsilon \cdot z)\;\;\;(W\cdot m^{-2})$$
where *A* is a parameter that depends on the thickness of the solidified shell, the size of the mould, and the thickness of the gaseous gap between the mould wall and the shell, *w_z_* is the casting speed (m·s^−1^), υ is an exponent obtained experimentally, characterizing the specific casting machine (m^−1^), and *z* is the coordinate in the casting direction (m).

However, the use of these types of formulas is not convenient, as they are usually tied to a certain type of the mould, the chemical composition of the cast steel, the height of the steel level in the mould and other technological parameters. When used with a different type of caster than for which they were compiled, they often achieve inaccurate results.

It is more precise, but more technically demanding, to determine the heat flux distribution in the mould experimentally using temperature probes in the mould wall.

Numerical models provide accurate solutions only if the boundary conditions and thermophysical parameters of the steel are precisely specified. Obtaining accurate boundary conditions is technically the most difficult phase of the modelling process.

Usually, the boundary condition in the mould is derived from the total heat flow into the cooling water, which is then distributed over the length of the mould according to an experimentally or theoretically determined function. The result of the modelling is usually the average thickness of the shell around the perimeter of the cross-section of the blank. But the real casting process is more complicated, as the steel does not solidify evenly around the perimeter [44]. The thickness of the shell in a given horizontal level varies around the perimeter by tens of percent. It follows from this fact that a breakout can occur even if the calculated average thickness of the shell at the end of the mould is sufficiently thick [36].

Due to the complexity of events in real casting conditions and the number of interdependent parameters, the use of mathematical methods would be unreliable without their tuning and verification according to real conditions. Close cooperation with the results of experimental measurements is always necessary as they introduce the characteristic features of a particular casting machine into the model [29]. It is not always possible to obtain the values of certain quantities on a real device that can be used to directly verify the results of the models: for example, the thickness of the solid shell. Measuring the shell thickness directly on the casting machine is basically impossible due to its technological and structural complexity and high temperatures.

The models are therefore tuned indirectly according to quantities that are easier to measure: usually according to temperatures in the mould walls or surface temperatures of the blank below the mould.

The kinetics of the temperature field of the mould wall generally carries the greatest amount of information about the complex process of solidification. The measured temperature field is evaluated quantitatively in terms of the magnitude of temperatures, qualitatively with regard to the symmetry of cooling, and further from the point of view of its dynamics, i.e., changes in the symmetry of heat removal over time and space and temperature fluctuations [38]. Large temperature fluctuations can indicate an uneven thickness of the shell, an uneven layer of casting powder or imperfect lubrication.

In exceptional cases, such as the one described below in this paper, it is possible to obtain a real shell after a breakout, against which the calculated shell thickness can be directly and accurately verified as well as the model algorithm itself.

### 2.2. Methodology of Simulation and Model Verification

The program was created at the authors’ workplace. The model is based on the method of discretization of the Fourier–Kirchhoff differential Equation (7) using the finite difference method. Given that the core of the calculation is the explicit method, the algorithm must check the numerical stability during the calculation, which represents the interdependence between the fineness of the computational mesh and the time step of the calculation.

The program includes an extensive database of information on steel chemical compositions, including their thermophysical properties (density, specific heat capacity, thermal conductivity coefficient) depending on temperature.

The temperature field during cooling and solidification is calculated at nodal points of the virtual blank. The dimensions of the mould, liquid steel temperature and chemical composition and casting parameters enter the model. For the primary cooling zone, the parameters are the inlet and outlet cooling water temperatures, the volume flow of the cooling water and the height of the steel level to calculate thermal boundary conditions. The advantage is when the wall of the mould is equipped with temperature sensors. Based on the chemical composition of the cast steel, the software defines thermophysical parameters, liquidus and solidus temperature. In the secondary cooling zone, the lengths of the individual subzones, the positions of the cooling nozzles and the heat transfer coefficients for each cooling nozzle are defined in the model [45].

The outputs of the model are numerical values and a graphic visualization of the results showing the course of temperatures of the surface and centre of the block depending on the cast length, temperature and phase maps in the longitudinal section of the block with the representation of isoliquidus and isosolidus boundaries. Additional results are metallurgical length, liquid core length and average shell thickness. The program allows a user to choose three characteristic points of the blank and monitor their temperatures graphically and numerically. The location of these positions is important for the reverse indirect verification of the model by comparing the calculated and measured surface temperatures of the block in the real caster (Figure 1) at particular positions. The program has been verified in this way during the casting of many heats.

Although the model calculates the complete temperature field of the blank in the caster, current research is focused on modelling the shell thickness at the exit of the mould. Using the numerical model, the growth of the thickness of the solid shell has been modelled as a function on the position in the mould for different values of the casting speed. The intensity of heat removal must input the model as a boundary condition. Since the casting speed and the heat removal from the mould are related, it is not possible to simply change only one of these quantities to obtain the dependence of the shell thickness on the selected quantity. The interdependence of these quantities is very complex, because the intensity of heat removal is affected also by shell shrinkage, the geometrical profile of the mould, lubricant transport into the gap, etc. The effort to comprehensively model these interactions is always too far from reality. Therefore, measured data from a real casting machine have been used as a boundary condition.

To determine the thermal boundary condition in the mould, it is necessary to measure the liquid steel temperature, the temperature increase and flow rate of the cooling water in the mould and several temperatures in the mould wall along its length. From these values, heat flux distribution along the mould can be derived. Usually in praxis, it is difficult to obtain a more detailed boundary condition, e.g., heat flux around the mould perimeter, so numerical models usually calculate the shell thickness only as a dependence on the longitudinal coordinate, which is averaged around the perimeter of the mould cross-section [44]. The liquid steel temperature is assigned to nodal points at the steel surface in the mould. Temperatures in the rest of nodal points are then calculated by the model.

The chemical composition of modelled steel was close to the average composition according to Table 1 [46]:

From the operational database, five time periods have been selected while the casting speed was constant at 0.38, 0.4, 0.47, 0.53 and 0.57 m·min^−1^. The steel level in the mould was 50% of the mould volume, the liquid steel temperature was close to the casting temperature 1542 °C, which means superheating 34 °C above liquidus temperature, the ambient temperature was 25 °C, the volume flow of cooling water in the mould was 120 m^3^·h^−1^, the temperature difference of the cooling water at the inlet and outlet of the mould was approximately 4.4 °C, the mould had a diameter of 410 mm and the length was 600 mm.

## 3. Results

A numerical thermal model has been created and implemented in a real caster to simulate the thickness of the solid shell. The model was verified indirectly according to the blank surface temperatures measured in the caster and directly by comparing the calculated shell thickness with the shell after the breakout.

### 3.1. Modelling of the Average Shell Thickness on the Real Caster

The total average heat flow has been evaluated from cooling water temperatures and flow rate (Figure 2). The mould was measured by temperature sensors six distances from its upper edge. The average temperature profiles in the mould wall during the selected time periods were evaluated (Figure 3). The temperature profile at the casting speed of 0.53 m/min is shifted due to the lower temperature of the inlet cooling water. Unlike the magnitude of wall temperature, temperature differences between the wall and water temperature are important as they are proportional to heat flux at the corresponding position. Using mould–water temperature differences, heat flux distributions in the mould were derived from total heat flows (Figure 4). Heat flux profiles were finally entered into the model as thermal boundary conditions.

The shell thickness calculated by the numerical model for the selected casting speeds is shown in Figure 5. Although the article focuses on the mould, an even greater length of the blank than the mould length was calculated. A thermal boundary condition below the mould was obtained by laboratory measurement of the heat transfer coefficient under cooling nozzles.

The numerical model was verified by comparison of the blank surface temperatures with temperatures measured by a real caster. However, in one case, it was possible to validate the model by direct comparing the calculated shell thickness with the actual shell obtained after breakout.

### 3.2. Analysis of the Shell Thickness of the Real Blank after Breakout

For the study of the growth of the solid shell, a part of the blank with a length of 1.7 m was used, which was left after a breakout during the continuous casting of a block with a round cross-section of a diameter measuring 410 mm.

Operating data have been recorded with a period of 5 s. Figure 6 shows a record of the casting speed, friction in the mould and the heat flow to the cooling water. On the horizontal axis is time relative to the moment of breakout. The casting speed was constant at 0.50 m∙min^−1^ for about 22 min before the breakout. In the time 280 s before the breakout, the prediction system indicated the danger of breakout based on the high value of friction in the mould. The caster control system responded by reducing the casting speed to 0.40 m∙min^−1^.

Friction in the mould was obtained from the monitoring and breakout prediction system DGS, which evaluates this quantity by an indirect method using an accelerometer and a pressure sensor in the hydraulics of the mould oscillation mechanism. Friction is expressed by a relative quantity called “the friction factor”.

Figure 7 shows the casting speed and the steel level, which is expressed by the distance of the steel surface from the upper edge of the mould. The moment the operator began again to increase the casting speed, a rapid drop in the steel level began, which was caused by the leakage of steel through a crack in the shell. During a further increasing in the casting speed in the next 20 s, the shell was exiting the mould while the liquid core was flowing out.

The cause of the breakout was a longitudinal crack on the side of the inner radius of the caster, which was more than 1 m long. The crack was caused by the shrinkage of the outer layer of the shell and thermal stress (Figure 8) in the temperature interval where the steel was of low strength. The formation of longitudinal cracks is more frequent in blanks of a circular cross-section than in rectangular ones, especially in casting peritectic steels, due to a large shrinkage.

For ease of handling, the shell was cut lengthwise into 10 parts, and each part was sequentially scanned using a 3D scanner and processed by a special software. From the data of the 3D model, it is possible to accurately determine the shape and size of the shell. A self-positioning and portable laser scanner HandyScan 3D EXAScan manufactured by Creaform was used. The camera makes it possible to capture objects with a higher resolution. The data acquire a dynamic resolution up to 0.05 mm with an accuracy of 40 μm∙m^−1^ dependent on the complexity of the shape and size of the object. A scanner uses a method where the surface of the scanned object is illuminated by a vibrating positional laser cross, which is then captured by two CCD cameras, and this image is subsequently evaluated using the triangulation method. This scanner is directly connected to a PC via a supplied cable, where real-time data processing takes place in one of the supported CAD/CAM systems that support the creation of a spatial network (e.g., SolidWorks).

The scanned objects were then modified by the software in a form that corresponds to the real state. A comparison of the scanned part of the shell with the real object is shown in Figure 9.

All parts of the shell were successively scanned and completed lengthwise (Figure 10). It can be seen from the figure that the crack originated in the mould on the side of the inner radius about 0.4 m from the steel surface and continued along the length of the block.

To determine the thickness of the shell, a surface network of points was created from the individual scanned parts of the shell. The mesh density can be set optionally. One part of the shell was divided into approximately 90,000 polygons (Figure 11 and Figure 12).

From the data obtained by scanning, the values of the thickness of the shell along its circumference and lengthwise with a constant angular and length step were determined. The side of the caster inner radius is assigned an angular coordinate of 0°, and the side of the outer radius corresponds to 180°. The formation of the so-called “flower” can be observed, i.e., the uneven growth of the thickness of the shell. Irregular growth of the shell is the result of uneven heat removal around the perimeter of the mould in conjunction with deformations caused by thermal stress.

Figure 13 shows the longitudinal profiles of the shell thickness at angular coordinates with a step of 45°.

The differences in the shell thickness around the perimeter increased with increasing distance from the meniscus. At the mould end, i.e., at the shell longitudinal position of approximately 0.55 m, the smallest shell thickness was 21.6 mm and the largest was 27.8 mm; at the mould end, the shell thickness varied from the average value by ±12.5%.

From the average values of the shell thickness around the perimeter of the cross-section, a regression function of the shell thickness as a function of time was derived
(17)$$\xi =1.131\cdot \tau^{0.679}\;\;\;(\mathrm{mm})$$

At the exit of the mould, the average thickness of the solid shell was 24.8 mm. The dependence confirms that even in this case, the parabolic law of growth of the solid shell does not apply.

### 3.3. Verification of a Numerical Model Using the Real Shell Thickness Measurement

The analysis of the geometry of the shell after the breakout was used to verify the thermal numerical model of the round block. Direct verification of the numerical model by comparing the calculated shell thickness with the measured thickness of the shell after the breakout is valuable, especially at the end of the mould where it is difficult to measure the blank surface temperatures due to intensive water spraying. Unfortunately, usually only limited breakouts are available.

The result of one such comparison is shown in Figure 14. At the end of the mould, the average thickness of the real shell was 24.8 mm, while the model calculated a thickness of 23.7 mm. The average relative error of the model along the mould length was less than ±2.5%.

Verification by shell thickness measurement confirmed that the model was well adjusted using previous indirect verification based on surface temperatures. The model is considered sufficiently accurate, and it is possible to assume that it can be applied for other boundary conditions as well.

## 4. Conclusions

Thermal processes in continuous steel casting significantly affect the quality of the cast production, and therefore, attention is paid to these processes. The cooling itself and heat removal in the mould and in the secondary zone are affected by a number of parameters that can be adjusted during casting. In this way, the optimal growth of the solid shell can be regulated, the resulting quality of the product can be increased and the formation of defects and cracks can be eliminated. The shell thickness is difficult to measure during real casting, so it is usually predicted by analytical, empirical or numerical models. Verification of these models is mostly performed indirectly using blank surface temperature measurements. The results of the models can be directly verified by comparison with the thickness of the shell obtained from the plant after the breakout. The developed methodology of shell geometry measurement and model verification for the round format with a diameter of 410 mm is described in the article. The same methodology can be applied for other shapes and sizes of cast profiles (square, rectangular). In these cases, the boundary conditions for different setups of the casting machine (mould, shape, size, etc.) must be defined based on experimental measurements.

This paper is focused on the description of the shell formation of a round block with a diameter of 410 mm. For experimental research, a part of the solid shell was obtained after the breakout. The 3D laser-scanning method was used to determine the thickness of the shell. The results show a power-law dependence of the average shell thickness on time, which differs from the often used parabolic dependence, as well as a significant non-uniformity of the shell thickness around the perimeter of the cross-section. At the end of the mould, the thickness around the perimeter fluctuates relatively by ±12.5%, which is absolutely in the range from 21.6 to 27.8 mm. The typical irregular shape, called a “flower”, which is often created during the casting of round profiles, was observed.

It can be stated that the casting of a circular cross-section is one of the most demanding process, especially in the case of steels with low carbon content. The round profile is resistant to changes in shape in the cross-section. Due to the shrinkage of the shell, the circular shape is therefore subjected to the much greater mechanical stress than rectangular profiles and faces a higher susceptibility to the formation of longitudinal cracks.

Using the geometric data of the actual shell thickness of the round block, the original simulation program was verified. The software is used to predict the temperatures, shell thickness and metallurgical length of round blanks. By comparing the modelled values, sufficient accuracy of the model for use in real casting conditions was demonstrated. According to the measured shell thickness profile, the time-dependent regression model for the given mould format and the particular casting machine was refined.

Retrieving a post-breakout shell from a factory is usually difficult, as the number of breakouts on modern casting machines equipped with predictive systems is low, and factories usually do not disclose this information. For these reasons, it is always valuable to be able to compare model results with physical reality. Other methods of validating numerical models, e.g., according to surface temperatures, are indeed used, but they are only indirect methods. The results presented in the article can also serve as valuable accompanying information for research on similar casting machines.

## Figures and Tables

**Figure 1 materials-16-05302-f001:**
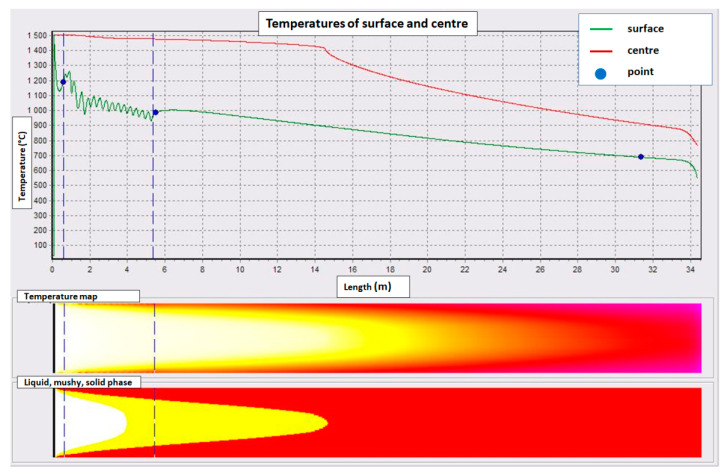
Example of modelled temperature field and phase composition map along the blank.

**Figure 2 materials-16-05302-f002:**
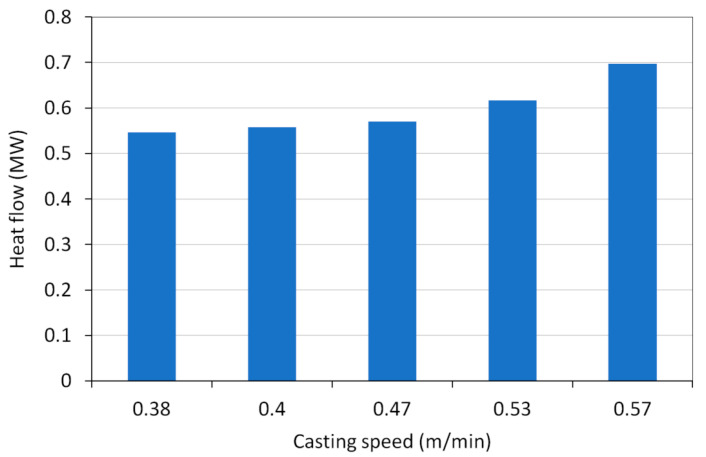
Heat flow of selected casting periods with different casting speed.

**Figure 3 materials-16-05302-f003:**
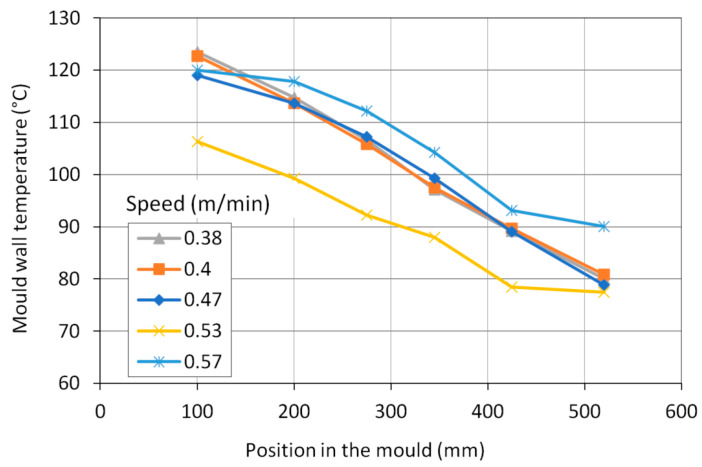
Average profiles of temperatures measured in the mould wall.

**Figure 4 materials-16-05302-f004:**
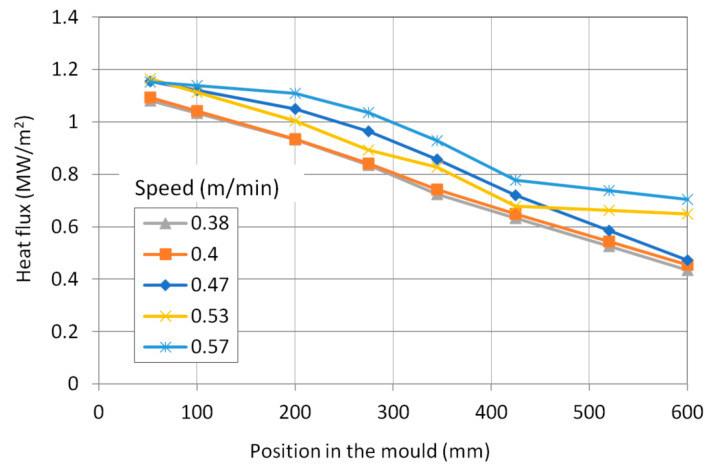
Average heat flux longitudinal profile during selected casting periods.

**Figure 5 materials-16-05302-f005:**
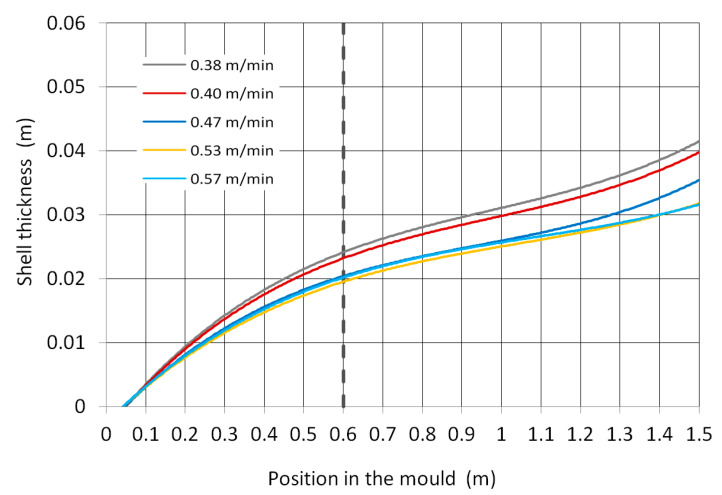
Shell thickness calculated by the numerical model.

**Figure 6 materials-16-05302-f006:**
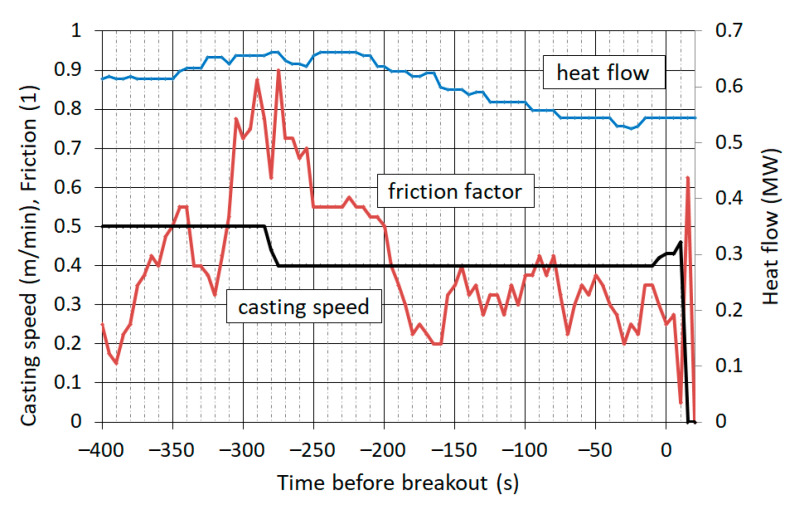
Recording of casting speed, friction factor and heat flow.

**Figure 7 materials-16-05302-f007:**
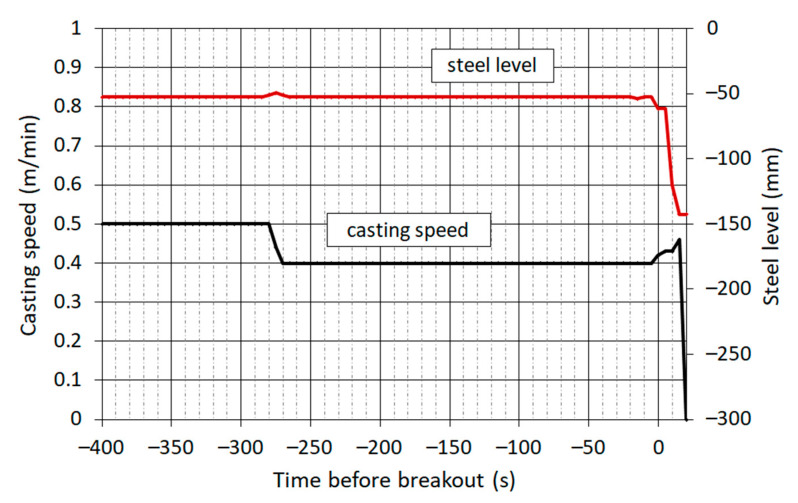
Recording of casting speed and position of steel level.

**Figure 8 materials-16-05302-f008:**
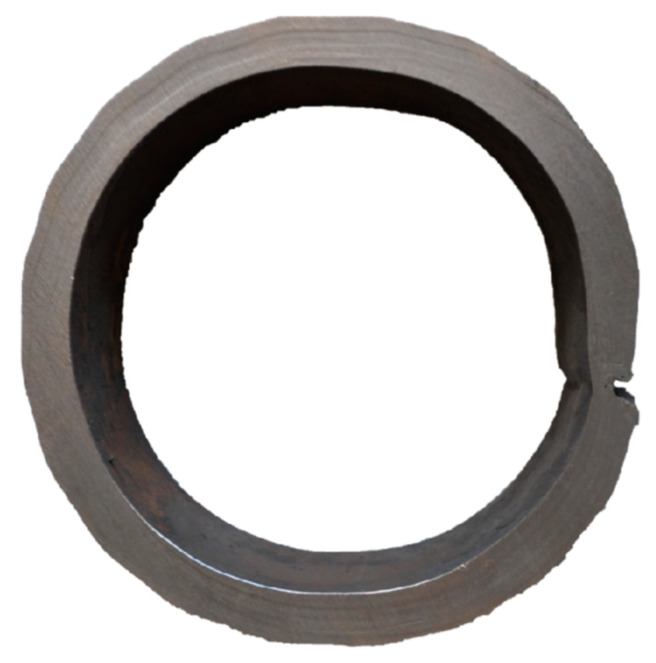
Cross-section of the shell with a crack.

**Figure 9 materials-16-05302-f009:**
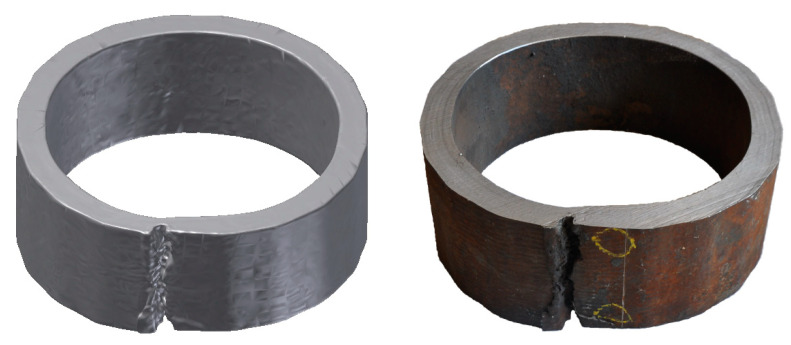
Scanned (**left**) and real (**right**) part of the shell.

**Figure 10 materials-16-05302-f010:**
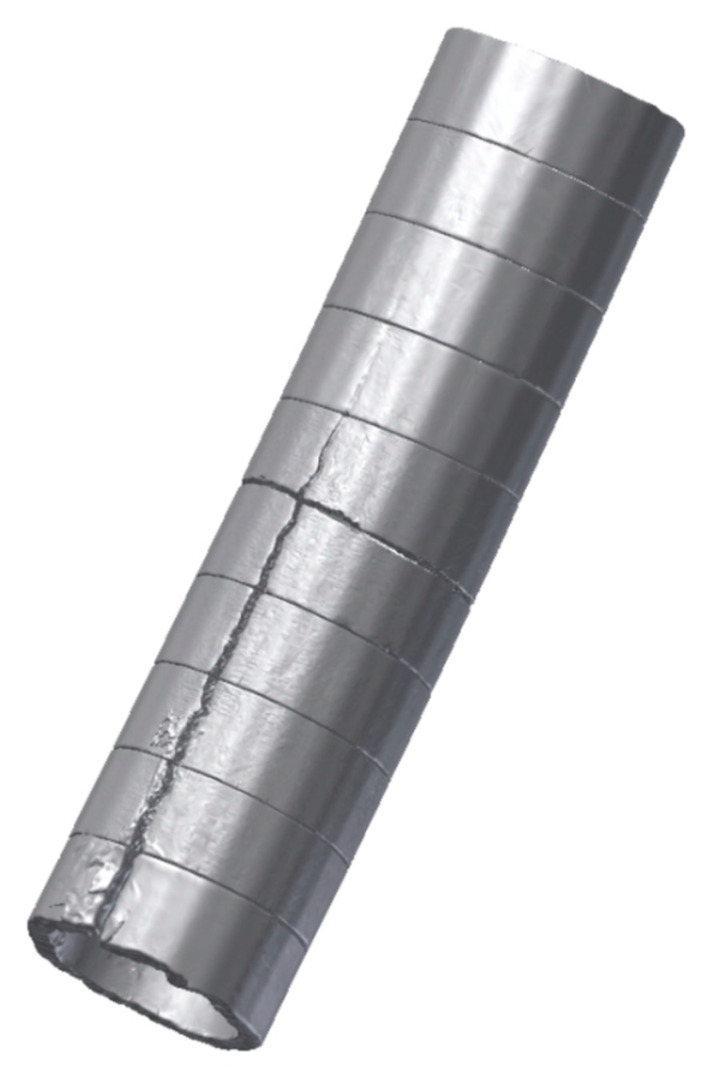
Scanned part of the shell.

**Figure 11 materials-16-05302-f011:**
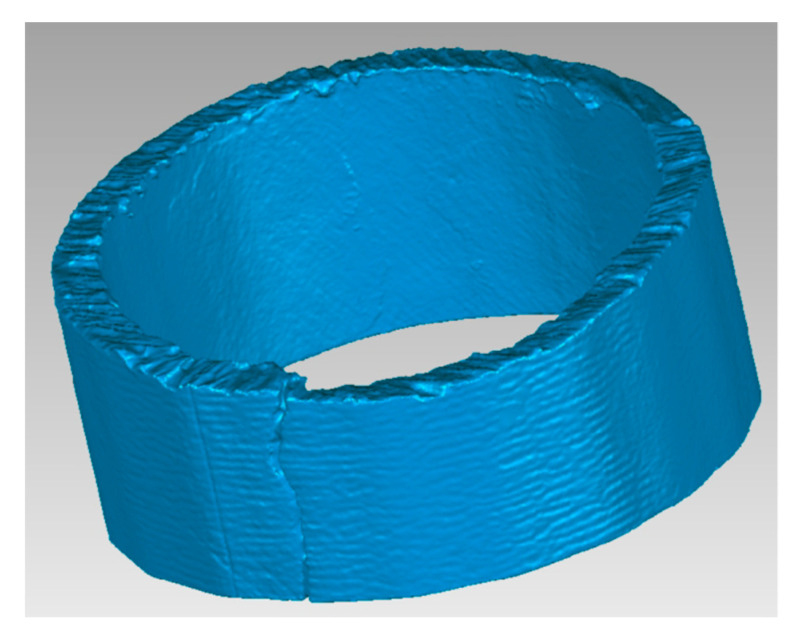
Scanned shell of the part of the block.

**Figure 12 materials-16-05302-f012:**
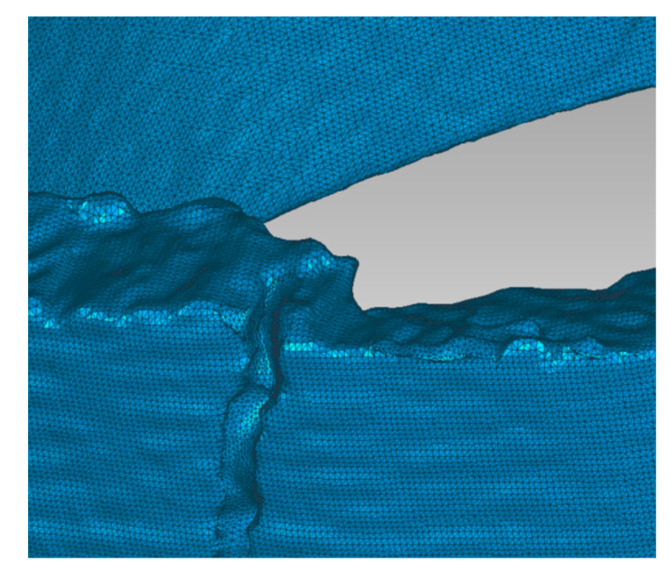
Detail of the model mesh.

**Figure 13 materials-16-05302-f013:**
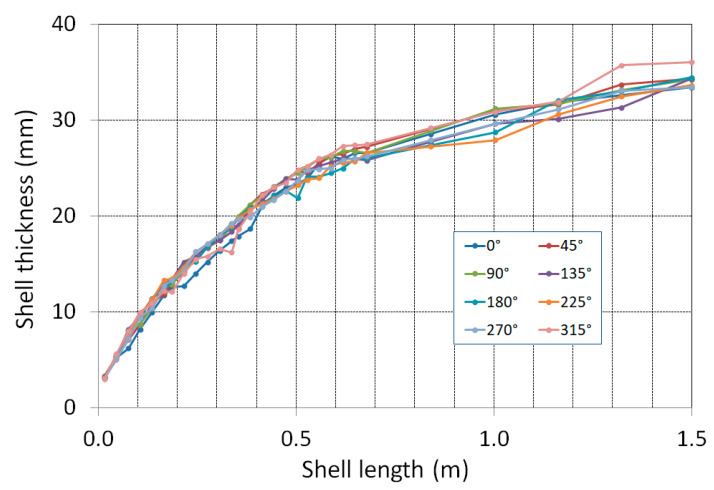
Measured thickness of the shell depending on the length.

**Figure 14 materials-16-05302-f014:**
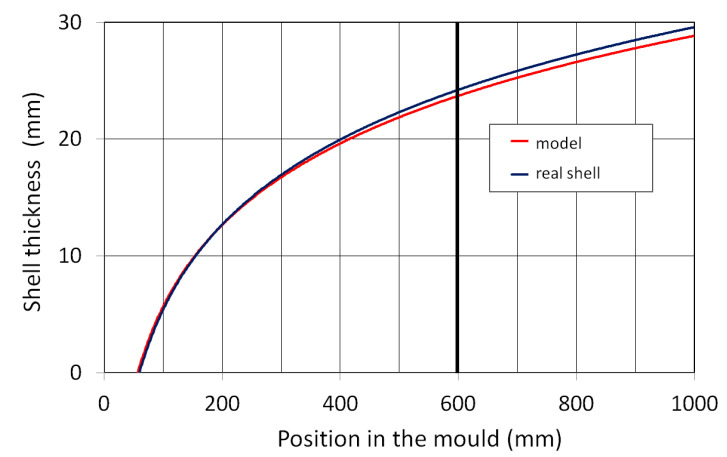
Comparison of calculated and measured shell thickness.

**Table 1 materials-16-05302-t001:** Steel chemical composition.

C	Mn	Si	P	S
0.168%	1.360%	0.390%	0.026%	0.024%

## Data Availability

The data presented in this study are available on request from the corresponding author.

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
