# Peer review of "Research on Solid Shell Growth during Continuous Steel Casting"

_materials, 2023, doi:10.3390/ma16155302_

Round 1
Reviewer 1 Report (Previous Reviewer 2)
The authors of the publication "Research on solid shell growth during continuous steel casting" presented the problem of solid shell growth and the measurement of the geometry of the real shell after the breakout, which occurred during the casting of the round block with a diameter of 410 mm from low carbon steel. The authors developed an original methodology was developed, in which a surface mesh of points was created from the individual scanned parts of the block using a 3D laser scanne. A three-dimensional model of the shell was subsequently created, from which shell thickness values along the blank length and around the cross sections were obtained.
Each of the presented parts of the publication has been correctly described by the authors. The conclusions are consistent and closely related to the research topic. The undoubted strength of the reviewed article is its rich description of the state of the issue with 34 references , including recent publications within the last five years.
As a reviewer, I think the work is good. It would be good to complete the data regarding the 3D scanner (scanner model).
Author Response
Please see the attachment.

Reviewer 2 Report (New Reviewer)
Dear authors:
I hope all of you are fine. My review is summarize as follows,
In this work quality disturbances affecting the continuous casting of steel consisting in the initiation and propagation of a crack process, termed as breakout, was investigated.
For such a purpose, a 3D digital model mesh of a breakout-affected-specimen was used to characterize morphological features and generate a numerical model to simulate cooling and solidification that takes place during the casting process. The 3D digital model mesh was obtained from a laser scanning process applied to a discarded defective piece of steel block cast having an annular cross section. The convenience of using this methodology is discussed together with the results.
This manuscript is interesting and deserve to be published. Previusly some problems have to be corrected:
a) A methodology section is missing. The reader must navigate throughout different sections of the manuscript to find out details of the applied methodology
b) The input temperature field considered in the thermal model is unclear.
c) Other
remarks are indicated in the attached pdf manuscript
file as pop-up messages.
Sincerely

I general the manuscript is well written there are some minor mistakes as mispelled words.
Author Response
Please see the attachment.

Reviewer 3 Report (New Reviewer)
The topic of the manuscript can be interesting to readers of Materials but revisions are required before consideration for publication.
1. The following published paper is recommended to be cited as they provide the latest review on the parameters that need to be optimized for the continuous casting process:
-Fabrication of aluminum metal matrix composite through continuous casting route: A review and future directions
2. In Line 46: “an extreme case, breaking the shell and leaking liquid steel, which is called breakout”. The following published paper is recommended to be cited as a reference as the paper gives readers a better understanding of the mould used in a continuous casting process:
=Investigation of failure and damages on a continuous casting copper mould
3. A couple of the cited literature are quite old references (more than 10 years ago). The authors are advised to cite more recent and updated references whenever possible.
4. The conclusion should be rewritten to highlight the novelty of this work.
Quality of English is reasonable.
Author Response
Please see the attachment.

Reviewer 4 Report (New Reviewer)
The presented article deals with the current issue, and after studying it, it is possible to assess that it is a research article.
- The overview of publications related to the issue of research into emerging solid shells is minimal,
- in the text under relation 2, it is defined that K is the solidification coefficient, while it is necessary to check the correctness of the units,
- under figure 2, the procedure of the experiment is described, where it would be appropriate to indicate which specific 3D scanner will be used and in which specific software the measured data will be processed,
- the text below Figure 4 states that the network density can be adjusted. Based on what 9000 polygons were set, is that enough to monitor such details?
- Figure 7 does not show anything in detail or anything specific. It would be advisable to replace it with a more suitable type of image or at least dimension the individual thicknesses of the shells.
- In the description under Figure 8, it would be appropriate to indicate the time in the given dependency. Is this a cool down time?
- Does Figure 9 show theoretical or real model temperatures?
Due to the fact that nowadays casting technology is common in practice and related to it various solutions for mold cooling, only a minimal part of the article is devoted to cooling. The results of the presented research are very simply described without concrete applications in practice for various other shapes and dimensions of the components. Based on the assessment, I recommend that the authors make minor changes to the article.
Author Response
Please see the attachment.

This manuscript is a resubmission of an earlier submission. The following is a list of the peer review reports and author responses from that submission.
Round 1
Reviewer 1 Report
1) The abstract is too long.
2) There are too many keywords.
3) In introduction section, it needs to be added more literature studies related to the investigation on solid shell growth during continuous steel casting.
4) The literature review is too general. It needs to discuss the literature gap in introduction section, the deeper discussion of the obtained literature is necessary, in order to show the really contribution to this research area, when compared with the existent and studied literature.
5) In a current form, a novelty of the conducted research is unclear.
6) In present article, it needs to add the differences and main research contributions of the present research compared with the existing research.
7) It needs to clearly indicate what software was used for performing the modelling of the solid shell growth. Furthermore, why this software was selected for the the modelling of the solid shell growth?
8) It is necessary to add more numerical model simulation details in the present article.
9) Similarly, it is necessary to add more experimental details in the present article.
10) The author wrote that “The result of one such comparison is shown in Fig. 10. At the end of the mould, the average thickness of the real shell was 24.8mm, the model calculated a thickness of 23.7mm. The relative error of the model was less than 2.5 %.”
Hence, it is necessary to provide the more reasonable and adequate explanation for the above research findings.
11) It is necessary to compare the differences between all the experimental and simulation results. Moreover, it needs to provide the more reasonable and adequate explanation for the differences between all the experimental and simulation results.
12) The abstract and conclusion sections need to be improved.
13) The results are mainly presented by figures. It is necessary to provide the more sufficient explanation of the research results. This is also the weakest aspect of the study.
14) The author needs to discuss all the research results in more detail and give more reasonable and sufficient explanation for all the research findings.
15) Results and discussion should be modified accordingly to the aim of the article.
16) The limitations of the study are not considered.
Moderate editing of English language required.
Reviewer 2 Report
The authors of the publication "Research on solid shell growth during continuous steel casting" presented the problem of solid shell growth and the measurement of the geometry of the real shell after the breakout, which occurred during the casting of the round block with a diameter of 410 mm from low carbon steel. The authors developed an original methodology was developed, in which a surface mesh of points was created from the individual scanned parts of the block using a 3D laser scanne. A three-dimensional model of the shell was subsequently created, from which shell thickness values along the blank length and around the cross sections were obtained.
Each of the presented parts of the publication has been correctly described by the authors. The conclusions are consistent and closely related to the research topic. The undoubted strength of the reviewed article is its rich description of the state of the issue with 34 references , including recent publications within the last five years.
As a reviewer of this work, however, I believe that the reviewed work requires many corrections, which will undoubtedly improve its quality:
1. The purpose of the work should be clearly defined.
2. Line 252 - "The operator responded..." - please write impersonally.
3. Line 270 - "...using a 3D scanner and processed by software." - What 3D scanner was used and what software. Please write more about this.
4. Figure 7. - I don't think this Figure is necessary. Unless you add dimensions to the Figure.
5. I miss in the article a discussion.